# Analytic approaches to clinical validation of results from preclinical models of glioblastoma: A systematic review

Beth Fitt[1⚬], Grace Loy[1⚬], Edward Christopher[2], Paul M. Brennan[3,4,5], Michael Tin Chung Poon[3,4,5,6¤]*

1 Edinburgh Medical School, University of Edinburgh, Edinburgh, United Kingdom, 2 Royal Infirmary of Edinburgh, NHS Lothian, Edinburgh, United Kingdom, 3 Brain Tumour Centre of Excellence, Cancer Research UK Edinburgh Centre, Edinburgh, United Kingdom, 4 Translational Neurosurgery, Centre for Clinical Brain Sciences, University of Edinburgh, Edinburgh, United Kingdom, 5 Department of Clinical Neuroscience, Royal Infirmary of Edinburgh, Edinburgh, United Kingdom, 6 Centre for Medical Informatics, Usher Institute, University of Edinburgh, Edinburgh, United Kingdom

⚬ These authors contributed equally to this work.
¤ Current address: Centre for Medical Informatics, Edinburgh, United Kingdom
* michael.poon@ed.ac.uk

Data Availability Statement: All data is available from the included studies.

Funding: Michael T.C. Poon is supported by Cancer Research UK Brain Tumour Centre of

## Abstract

### Introduction

Analytic approaches to clinical validation of results from preclinical models are important in assessment of their relevance to human disease. This systematic review examined consistency in reporting of glioblastoma cohorts from The Cancer Genome Atlas (TCGA) or Chinese Glioma Genome Atlas (CGGA) and assessed whether studies included patient characteristics in their survival analyses.

### Methods

We searched Embase and Medline on 02Feb21 for studies using preclinical models of glioblastoma published after Jan2008 that used data from TCGA or CGGA to validate the association between at least one molecular marker and overall survival in adult patients with glioblastoma. Main data items included cohort characteristics, statistical significance of the survival analysis, and model covariates.

### Results

There were 58 eligible studies from 1,751 non-duplicate records investigating 126 individual molecular markers. In 14 studies published between 2017 and 2020 using TCGA RNA microarray data that should have the same cohort, the median number of patients was 464.5 (interquartile range 220.5–525). Of the 15 molecular markers that underwent more than one univariable or multivariable survival analyses, five had discrepancies between studies. Covariates used in the 17 studies that used multivariable survival analyses were age (76.5%), pre-operative functional status (35.3%), sex (29.4%) MGMT promoter methylation (29.4%), radiotherapy (23.5%), chemotherapy (17.6%), IDH mutation (17.6%) and extent of resection (5.9%).

Excellence Award (C157/A27589). The funder had no role in study design, data collection and analysis, decision to publish, or preparation of the manuscript.

**Competing interests:** The authors have declared that no competing interests exist.

## Conclusion

Preclinical glioblastoma studies that used TCGA for validation did not provide sufficient information about their cohort selection and there were inconsistent results. Transparency in reporting and the use of analytic approaches that adjust for clinical variables can improve the reproducibility between studies.

## Introduction

Glioblastoma, the most common primary brain cancer, is a fatal disease with patients' median survival of 6–8 months [1,2]. Novel therapies from translational research are desperately needed because current therapeutic options have only a modest and temporary impact on survival [3,4]. Discovery science has advanced our understanding of cancer cell biology and is a step towards developing novel therapies [5]. These discoveries are usually based on preclinical models, from which the relevance to human disease must be established. Demonstrating relevance requires quality clinical and biological data. The Cancer Genome Atlas (TCGA) [6] and the Chinese Glioma Genome Atlas (CGGA) [7] are two open-access resources from which laboratory scientists can interrogate human data to verify their findings in preclinical glioblastoma research. These resources are valuable for the molecular characterisation of glioblastoma and can also be used to examine the associations between molecular markers of interest and survival. An association with survival might implicate a molecular marker as a potential drug target.

Survival analyses using only genomic data are unlikely to have adequate clinical relevance because clinical factors also affect survival. An imbalance of clinical characteristics between comparison groups can confound the association between the molecular marker and survival. Univariable survival analyses that take on only one molecular marker do not account for other markers or clinical characteristics [8]. The resulting associations from such analyses are subjected to confounding effects, which may render them unreliable. Confounding is a fundamental issue that affects observational health-related research, and it should be controlled for when possible [9]. Multivariable analyses are methods to control for confounders and are, therefore, preferable. Open access policies for data and code sharing should facilitate the re-use of data and reproducibility of results [10]. Transparent and detailed reporting of the analytic approach is crucial for replicability and comparison of analyses. These methodological aspects can ensure the science that progresses to clinical trials is well-founded.

Clinical validation of results from preclinical glioblastoma studies using TCGA or CGGA data represents a common experimental step to substantiate research findings. This systematic review examined these studies for their consistency in reporting of cohorts from TCGA and CGGA and whether they included patient characteristics in their survival analyses.

## Methods

### Eligibility criteria

This review included studies that used data from TCGA or CGGA to examine the association between at least one molecular marker and overall survival in adult patients aged ≥18 years diagnosed with non-recurrent histopathologically confirmed glioblastoma. Studies using any molecular data type from TCGA or CGGA were eligible. Studies using both TCGA and CGGA were eligible if they had separately reported results for TCGA and CGGA. We only

included studies that used cell or animal models to first identify molecular markers associated with tumour biology, then examined the association between these markers and overall survival in humans using TCGA or CGGA data. We excluded case reports, reviews, editorials and conference abstracts (S1 File).

## Study selection

We searched Embase and Medline on 02 February 2021 for potentially eligible studies published after January 2008 using search terms relating to "glioma", "survival", "TCGA" and "CGGA" (S2 File). The lower limit of the search period was set because data from TCGA first became available in 2008. After removing duplicate studies, two independent reviewers (B.F. and G.L.) performed screening using titles and abstracts followed by full-text eligibility assessment. Any disagreements at each stage were resolved through discussion with a third reviewer (M.T.C.P.).

## Data extraction and data items

Two reviewers (B.F. and G.L.) independently collected data from each study using the online systematic review management software Covidence (Veritas Health Innovation, Melbourne, Australia. Available at www.covidence.org). Disagreements were resolved by discussion between the two reviewers or by involving a third reviewer (M.T.C.P.). Data items included study characteristics, TCGA cohort characteristics, CGGA cohort characteristics, genomic data used, molecular markers, and details of survival analysis. Molecular markers included expression, variants, or methylation of genes, RNAs and microRNAs. A set of molecular markers was defined by the analysis of >1 molecular markers together.

Each study can report results from multiple survival analyses using the overall cohort or specific subgroups (S1 Fig). We collected information on all survival analyses performed in the studies. We categorised survival analysis into univariable and multivariable analysis, and we collected the covariates entered into the multivariable analysis. To describe the association between molecular markers and survival, we considered the reported p value of <0.05 as statistical significance. If a study reported results from both TCGA and CGGA cohorts, we extracted the statistical significance of these results separately. Data on effect sizes and their corresponding 95% confidence intervals (CI) were not collected because studies using log-rank (Mantel-Cox) tests to compare survival between study-specific groups do not provide these data and there was no plan for meta-analysis.

## Quality assessment

There was no risk of bias assessment tool directly relevant to studies in this review. However, we assessed components of the study design relating to risk of bias. These measures of quality included types and size of cohorts used for survival analysis, types of genomic data used from TCGA or CGGA, and the criteria used to select patients for survival analysis. We did not quantify the quality of study based on risk of bias items because this review aimed to assess the reporting and approach to analyses rather than to summarise effect sizes.

## Summary statistics

We presented study characteristics, results and quality measures using descriptive statistics with stratification by type of survival analysis, univariable and multivariable, where available. The availability of data in TCGA increased over time and there are different numbers of patients in whom various types of data are available. To assess the reproducibility of cohort

selection from TCGA, we summarised the number of patients in studies published between 2017–2020 using TCGA RNA microarray. These studies should have the same number of patients because they all used the same RNA microarray dataset from TCGA when there was no further accrual of patients. There were occasions when two or more survival analyses within or between studies investigated the association between a molecular marker and survival. We presented findings on these molecular markers that underwent two or more analyses to demonstrate the consistencies of results. There was no meta-analysis of any association between molecular markers and overall survival.

## Results

### Study characteristics

This review included 58 eligible studies from 1,751 non-duplicate records retrieved from our systematic search (Fig 1 and S1 References). Individual study characteristics are presented in S1 Table. These studies investigated 126 individual molecular markers and 32 sets of molecular markers. Most (62.1%) studies were published in 2017–2020 and were from research teams based in the United States (34.5%), China (27.6%) and Europe (24.1%). The pre-clinical glioblastoma models used were cell lines and orthotopic mouse models in 51.7% and 48.3% studies, respectively. All studies used a form of data from TCGA with various combination with other data sources and two studies used data from CGGA (Table 1). RNA microarray data was the most common data type, used in 45 (77.6%) studies. Three (5.2%) studies did not specify the data type used. Six studies (five using TCGA data and one using both TCGA and CGGA data) did not provide the number of patients included.

When investigating the association between their markers of interest from pre-clinical models and survival using genomic data, more studies used univariable survival analyses only (70.7%) compared to those that used multivariable analyses (29.3%). All univariable analyses used the non-parametric log-rank (Mantel-Cox) method and all multivariable analyses used the Cox proportional hazards regression. There were 16 (27.6%) studies that described additional criteria for patient inclusion within the selected TCGA cohort.

**Reproducibility and survival analysis.** The date and requested data type of query in TCGA can result in a different number of patients available for survival analysis. To assess reproducibility of cohort selection from TCGA in the included studies, we summarised the numbers of patients in studies with similar data specifications. In 14 studies published between 2017 and 2020 using TCGA RNA microarray data without additional patient inclusion criteria, the median number of patients included was 464.5 (interquartile range [IQR] 220.5–525). Of these studies, 12 studies did not perform a multivariable survival analysis, therefore all should have the same number of patients included; the median number of patients included in the univariable survival analysis was 467 (IQR 196.75–528.75).

Among the 126 distinct molecular markers investigated in the included studies, 15 markers underwent more than one univariable or multivariable survival analysis (Table 2). The association of these markers with outcomes were consistent between different analyses most of the time. However, there were discrepancies between results for C-X-C Motif Chemokine Ligan 14 (CXCL14), epidermal growth factor receptor (EGFR), netrin 4 (NTN4), SRY-Box transcription factor 2 (SOX2), serglycin (SRGN) and miRNA-17-5p microRNA (Table 2). These discrepancies appear to relate to the type of survival analysis used (CXCL14, SOX2, SRGN) or the data type (EGFR, NTN4).

There were 17 studies that investigated the association between their molecular markers of interest and overall survival using a multivariable survival analysis. All these studies used TCGA data, which have clinical data available. The most frequently included clinical variable

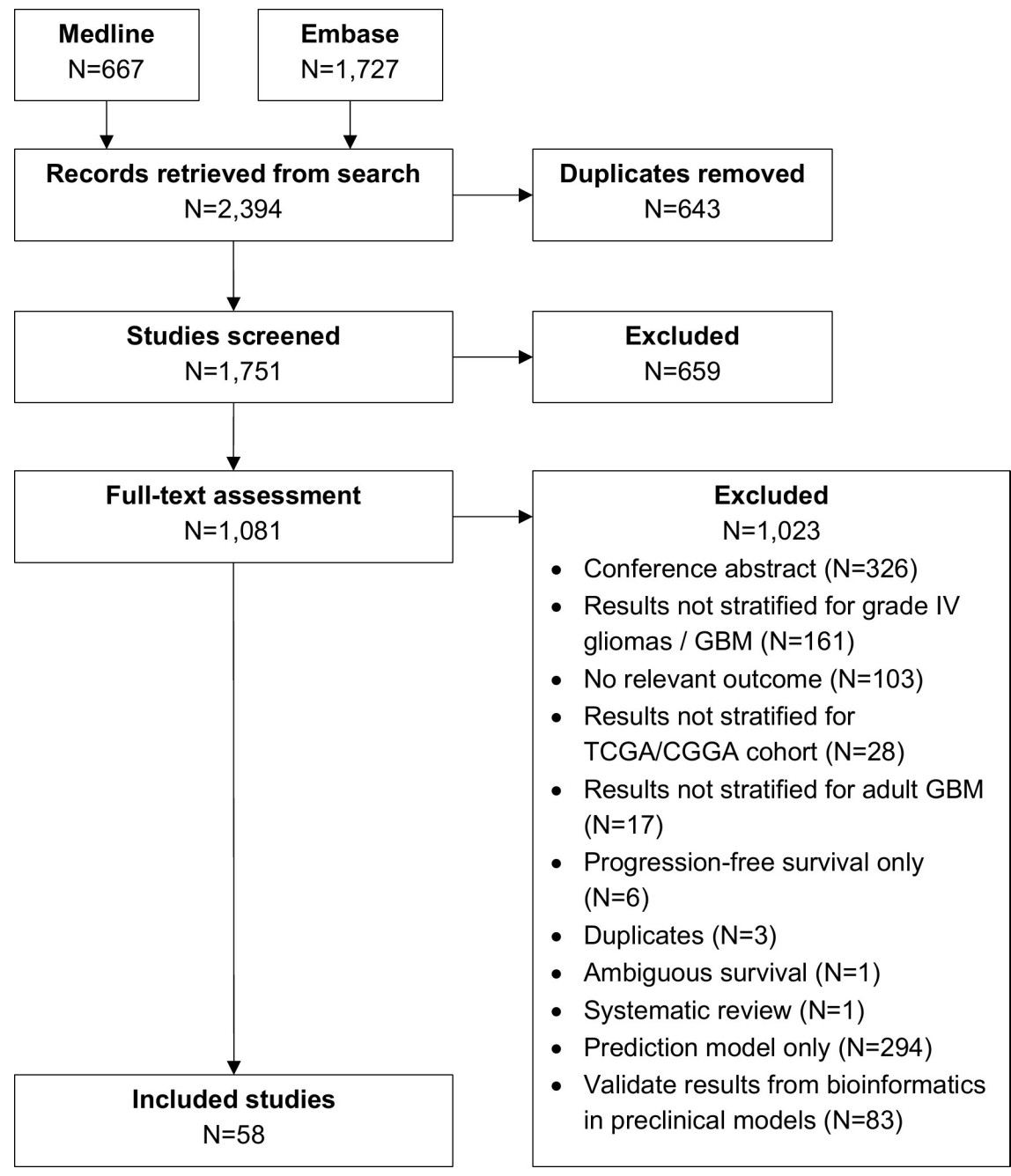

**Fig 1. PRISMA flowchart of study selection.**

in the multivariable model was age (76.5%) (Fig 2). Other variables included pre-operative functional status (35.3%), sex (29.4%), MGMT promoter methylation (29.4%), radiotherapy (23.5%), chemotherapy (17.6%), IDH mutation (17.6%) and extent of resection (5.9%).

## Discussion

There were studies in glioblastoma research that used data from publicly available genomic repositories to correlate pre-clinical experimental findings with clinical survival benefit in

**Table 1. Characteristics of 58 included studies that used TCGA or CGGA data to validate findings from experiments using pre-clinical models of glioblastoma.**

| | | Survival analysis type | |
|---|---|---|---|
| | Overall N = 58 | Univariable N = 41 | Multivariable N = 17 |
| **Year of publication** | | | |
| 2009–2012 | 4 (6.9%) | 2 (4.9%) | 2 (11.8%) |
| 2013–2016 | 18 (31.0%) | 13 (31.7%) | 5 (29.4%) |
| 2017–2020 | 36 (62.1%) | 26 (63.4%) | 10 (58.8%) |
| **Country / region** | | | |
| United States | 20 (34.5%) | 13 (31.7%) | 7 (41.2%) |
| Europe (inc. UK) | 14 (24.1%) | 9 (22.0%) | 5 (29.4%) |
| China | 16 (27.6%) | 13 (31.7%) | 3 (17.6%) |
| Other countries[a] | 8 (13.8%) | 6 (14.6%) | 2 (11.8%) |
| **Pre-clinical model** | | | |
| Cell lines | 30 (51.7%) | 24 (58.5%) | 6 (35.3%) |
| Orthotopic mouse models | 28 (48.3%) | 17 (41.5%) | 11 (64.7%) |
| **Data source** | | | |
| TCGA only | 34 (58.6%) | 26 (63.4%) | 8 (47.1%) |
| TCGA & CGGA | 1 (1.7%) | 0 (0.0%) | 1 (5.9%) |
| TCGA and other public sources | 9 (15.5%) | 6 (14.6%) | 3 (17.6%) |
| TCGA and own patients | 13 (22.4%) | 8 (19.5%) | 5 (29.4%) |
| TCGA, CGGA and other public sources | 1 (1.7%) | 1 (2.4%) | 0 (0.0%) |
| **Experimental strategy** | | | |
| RNA microarray only | 27 (46.6%) | 24 (58.5%) | 3 (17.6%) |
| RNA sequencing only | 7 (12.1%) | 4 (9.8%) | 3 (17.6%) |
| miRNA microarray only | 2 (3.4%) | 1 (2.4%) | 1 (5.9%) |
| RNA microarray and RNA sequencing | 4 (6.9%) | 3 (7.3%) | 1 (5.9%) |
| RNA microarray and miRNA microarray | 10 (17.2%) | 6 (14.6%) | 4 (23.5%) |
| RNA sequencing and miRNA microarray | 1 (1.7%) | 0 (0.0%) | 1 (5.9%) |
| RNA microarray and DNA methylation | 1 (1.7%) | 0 (0.0%) | 1 (5.9%) |
| RNA sequencing, RNA microarray and miRNA microarray | 2 (3.4%) | 0 (0.0%) | 2 (11.8%) |
| RNA sequencing, RNA microarray and DNA methylation | 1 (1.7%) | 0 (0.0%) | 1 (5.9%) |
| Unspecified | 3 (5.2%) | 3 (7.3%) | 0 (0.0%) |
| **Prognostic marker of interest** | | | |
| One marker only | 21 (36.2%) | 20 (48.8%) | 1 (5.9%) |
| >1 individual markers | 13 (22.4%) | 10 (24.4%) | 3 (17.6%) |
| Set(s) of markers only | 7 (12.1%) | 4 (9.8%) | 3 (17.6%) |
| One marker and set(s) of markers | 2 (3.4%) | 2 (4.9%) | 0 (0.0%) |
| >1 individual markers and set(s) of markers | 10 (17.2%) | 4 (9.8%) | 6 (35.3%) |
| One marker and sets of markers with clinical variable(s) | 4 (6.9%) | 1 (2.4%) | 3 (17.6%) |
| Sets of markers and markers with clinical variable(s) | 1 (1.7%) | 0 (0.0%) | 1 (5.9%) |

[a]Other countries included Brazil, Canada, India, Israel, Republic of Korea and Taiwan. UK = United Kingdom; TCGA = The Cancer Genome Atlas; CGGA = Chinese Glioma Genome Atlas; miRNA = micro-RNA.

humans. These studies often had different numbers of patients included despite using the same data source and data type. Survival analyses often did not include other critical clinical variables associated with survival such as extent of resection [11], chemotherapy and radiotherapy [3,12]. In studies that performed a multivariable survival analysis, most clinical variables such as extent of resection and oncological treatment were not included. This yielded

**Table 2. Results of molecular markers that were reported in two or more separate survival analyses.**

| Molecular marker | Consistency | Author | Data type | Analysis type | Direction of association |
|---|---|---|---|---|---|
| CXCL14 | No | Zeng 2018 | RNA-Seq, RNA microarray and miRNA microarray | U | Neg |
|  |  |  |  | M | - |
| EGFR | No | Kuang 2018 | RNA microarray only | U | Pos |
|  |  | Li 2018 | RNA-Seq only | U | - |
| HOTAIR | Yes | Xavier-Magalhaes 2018 | RNA-Seq, RNA microarray and DNA methylation | U | Neg |
|  |  |  |  | M | Neg |
| IDO1 | Yes | Zhai 2017 | RNA microarray and RNA-Seq | U | Neg |
|  |  |  |  | M | Neg |
| IL-8 | Yes | Hasan 2019 | RNA microarray only | U | Neg |
|  |  |  |  | M | Neg |
| MARCKS | Yes | Jarboe 2012 | RNA microarray and DNA methylation | U | Pos |
|  |  |  |  | M | Pos |
| miR-17-5p | No | Zeng 2018 | RNA-Seq, RNA microarray and miRNA microarray | U | Pos |
|  |  |  |  | M | - |
| miR-181d | Yes | Genovese 2012 | RNA microarray and miRNA microarray | U | - |
|  |  | Ho 2017 | RNA-Seq, RNA microarray and miRNA microarray | U | - |
| miR-34a | Yes | Genovese 2012 | RNA microarray and miRNA microarray | U | Neg |
|  |  |  |  | M | Neg |
| NTN4 | No | Hu 2012 | RNA microarray only | U | Pos |
|  |  | Li 2018 | RNA-Seq only | U | - |
| PD-L1 | Yes | Nduom 2016 | RNA-Seq only | U | Neg |
|  |  |  |  | M | Neg |
| POSTN | Yes | Mega 2020 | RNA microarray only | U | Neg |
|  |  | Liu 2019 | RNA microarray and miRNA microarray | U | Neg |
|  |  | Mega 2020 | RNA microarray only | M | Neg |
| SFRP1 | Yes | Delic 2014 | RNA microarray and miRNA microarray | U | Pos |
|  |  |  |  | M | Pos |
| Sox2 | No | Sathyan 2015 | RNA microarray and miRNA microarray | U | Pos |
|  |  |  |  | M | - |
| SRGN | No | Mega 2020 | RNA microarray only | U | Neg |
|  |  |  |  | M | - |

Consistency refers to the association between a molecular marker and survival being statistically significant in different analyses. Inconsistencies of associations with survival: Same analysis type and different data type (EGFR, NTN4) and different analysis type on same data type (CXCL14, miR-17-5p, Sox2, SRGN). Molecular markers ordered alphabetically. Full references available in Supplementary Materials. RNA-Seq = RNA sequencing; No = not consistent between different analyses; Yes = consistent between different analyses; U = univariable survival analysis; M = multivariable survival analysis; Pos = positive association i.e. higher levels of the molecular marker associated with better survival and $p<0.05$; Neg = negative association i.e. lower levels of molecular marker associated with worse survival and $p<0.05$;— = statistical significance not demonstrated ($p \geq 0.05$).

some inconsistent results between studies. Other results were subject to confounding effects by clinical variables that were not accounted for.

## Reproducibility

Research reproducibility encompasses several aspects: consistent results based on the same data and analysis, consistent results based on the same data but different analyses, consistent results from new data based on previous study design of another study, and consistent results from another study with a similar study design [13,14]. Our review addressed the first two of

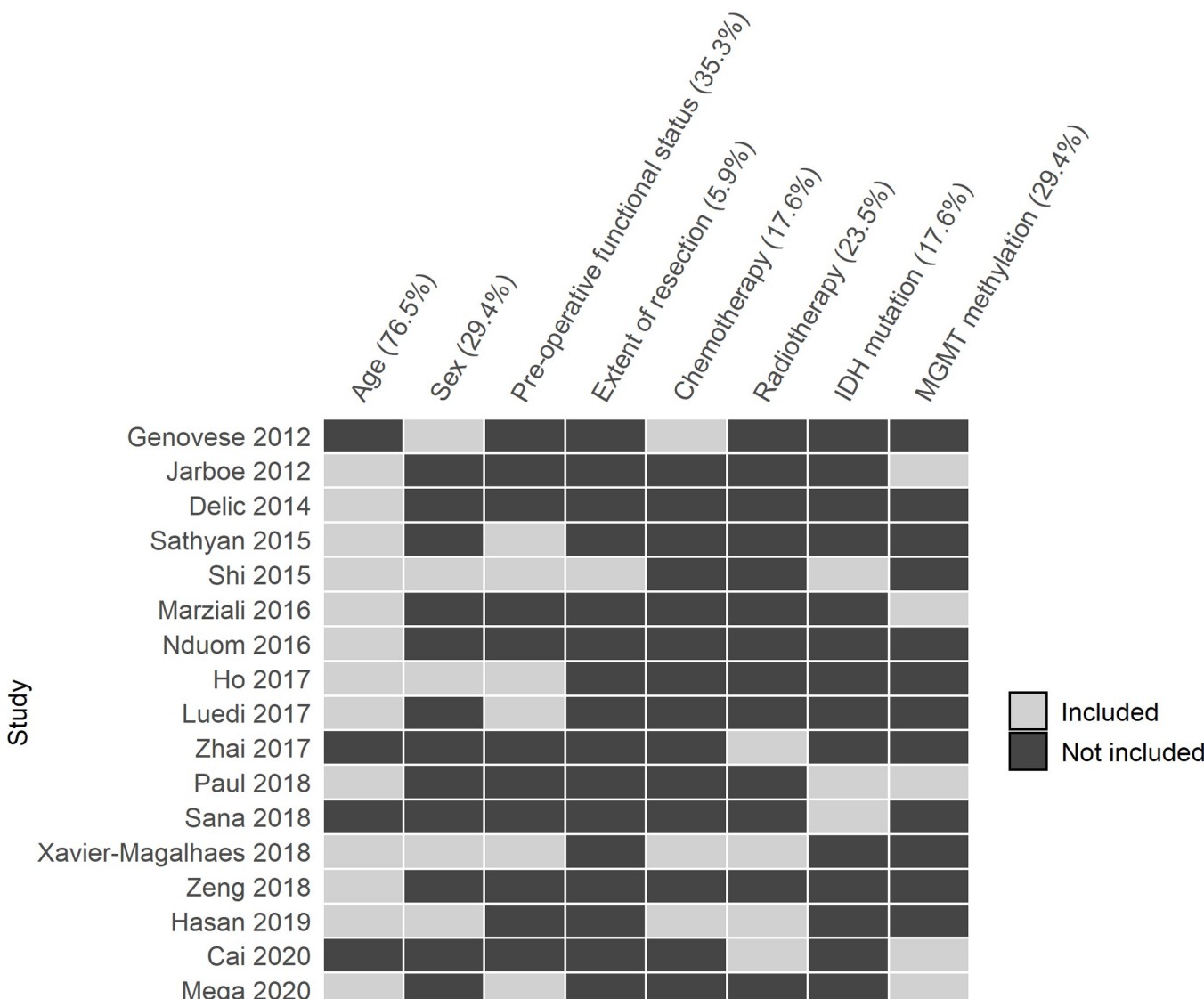

**Fig 2. Clinical variables entered analyses in 17 studies that used a multivariable survival model.** Rows represent studies that used a multivariable model for survival analysis (S1 References). Columns are clinical variables relevant to survival in patients with glioblastoma.

these aspects. Development of novel cancer therapies relies on reproducible results from preclinical research. The need for improving reproducibility is not new [15]. In cancer research, there is a heavy reliance on the preclinical literature for drug development [16]. However, issues with reporting bias, suboptimal reporting quality, varying reproducibility and preclinical model representation of disease impede the success in finding new therapies [17]. The availability of survival data in publicly available data from cancer genomics programmes presents an opportunity for researchers to assess the association between molecular markers and patient survival in a reproducible manner. These open access data sources provide data on the same cohort of patients, which encourages reproducibility between studies. However, our findings demonstrate that patient selection was not adequately described, resulting in different numbers of patients between studies that supposedly used the same dataset. There are reproducible ways of querying TCGA data, for example, using the 'TCGABiolinks' R/Bioconductor package [18] where code-based commands can be shared as supplementary materials.

Adopting relevant aspects of reporting guidelines such as Strengthening the Reporting of Observational Studies in Epidemiology (STROBE) [19], Transparent reporting of a multivariable prediction model for individual prognosis or diagnosis (TRIPOD) [20] and REporting recommendations for tumour MARKer prognostic studies (REMARK) [21] can further improve transparency in reporting.

## Confounding effects of clinical variables

Confounding is an important consideration in analysing observation data. A confounder can diminish or exaggerate the association between the exposure and the outcome, leading to spurious results [22]. Confounding effects may be controlled by design or by analysis—the latter is most relevant in this review. Control by analysis refers to adopting an analysis method that adjusts for confounders. There are many ways to achieve this, such as stratified and various regression models [9]. The most commonly used multivariable survival analysis is the Cox regression [8]. Most studies in this review did not consider clinical variables as potential confounders to the association between the molecular marker of interest and survival. There are nevertheless examples of associations that no longer exhibit a statistical significance after adjustment to clinical variables in a multivariable analysis (Table 2). Therefore, it is important to explore and consider confounders when assessing the effect of molecular markers on survival [23]. This is not a simple task because of data missingness, relatively small numbers of patients available, as well as correlations between clinical variables. Both data driven and clinically informed choice of covariates would be a reasonable approach [24].

## Strengths and limitations

This systematic review assessed all pre-clinical studies that used data from TCGA or CGGA to validate findings from their laboratory experiments. Our data collection allowed comparison of findings between and within studies, which allowed our evaluation of replicability.

Clinical studies that examined associations of previously investigated molecular markers with survival were not included in this review. These studies may provide more detailed descriptions of cohort selection and may be more likely to consider confounding effects from clinical variables. This would mean an overestimation of inconsistencies and suboptimal analytic approaches in our review. However, any omission of consideration about patients being more than their tumours should be highlighted to re-orientate research focus to patient benefits. Collecting data on p values only to denote statistical significance was a pragmatic approach to describing associations reported in the included studies, since most studies did not report any effect sizes. This does not represent our views on the appropriate statistical approach and reporting of findings. We advocate reporting of effect sizes with their corresponding precision, adjusting for confounders. P values should not be used as a cut-off for the significance of an association [25]. There are other aspects of survival analyses that we did not assess, such as whether included studies tested for the proportional hazard assumption when using a Cox regression [26]. While these analytic procedures are important, reporting of these would not affect our findings. We were unable to perform meta-analyses of the associations between molecular markers and survival because studies were not comparable and there were few effect sizes reported. This limitation prevented us from quantifying the consistency based on heterogeneity and variance measures.

## Conclusions

Translational studies in glioblastoma research should increase their transparency to facilitate replicability. The validation of laboratory experimental findings using human data is

important to demonstrate translational value; but this should be done with consideration of patient characteristics. Integration of expertise in pre-clinical, genomic and clinical studies may help to address the challenge of producing replicable and meaningful research through collaboration between scientists in different fields.

## Supporting information

**S1 Checklist.**
(DOCX)

**S1 Fig. Common analytic strategy used by included studies.**
(PDF)

**S1 Table. Characteristics of 58 included studies.** Data type: A = RNA microarray only, B = RNA microarray and miRNA microarray, C = RNA sequencing only, D = RNA microarray and RNA sequencing, E = miRNA microarray only, F = RNA sequencing, RNA microarray and miRNA microarray, G = RNA sequencing and miRNA microarray, H = RNA microarray and DNA methylation, I = RNA sequencing, RNA microarray and DNA methylation, J = Unspecified. If a study used a data source but not specified the number of patients, the column for data source would be "Yes [NS]" indicating number of patients not specified.
(PDF)

**S2 Table. References to specific analyses extracted for comparison of results on molecular markers.** Molecular markers ordered alphabetically accompanied with the location of analysis in the original manuscript. U = univariable survival analysis; M = multivariable survival analysis; ▲ = positive association i.e. higher levels of the molecular marker associated with better survival and $p < 0.05$; ▼ = negative association i.e. lower levels of molecular marker associated with worse survival and $p < 0.05$; □ = statistical significance not demonstrated ($p \geq 0.05$).
(PDF)

**S1 References. Full references of included studies.**
(PDF)

**S1 File. List of eligibility criteria.**
(PDF)

**S2 File. Search strategy in Medline and Embase.**
(PDF)

## Author Contributions

**Conceptualization:** Michael Tin Chung Poon.

**Data curation:** Beth Fitt, Grace Loy, Edward Christopher, Michael Tin Chung Poon.

**Formal analysis:** Beth Fitt, Grace Loy.

**Investigation:** Grace Loy, Michael Tin Chung Poon.

**Methodology:** Beth Fitt, Grace Loy, Michael Tin Chung Poon.

**Project administration:** Beth Fitt, Grace Loy.

**Supervision:** Edward Christopher, Paul M. Brennan.

**Visualization:** Michael Tin Chung Poon.

**Writing – original draft:** Beth Fitt, Grace Loy, Michael Tin Chung Poon.

**Writing – review & editing:** Edward Christopher, Paul M. Brennan, Michael Tin Chung Poon.

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
