## [Decision Letter · Decision Letter 0]

10 Dec 2021

PONE-D-21-32960Analytic approaches to clinical validation of results from preclinical models of glioblastoma: a systematic reviewPLOS ONE

Dear Dr. Poon,

Thank you for submitting your manuscript to PLOS ONE. After careful consideration, we feel that it has merit but does not fully meet PLOS ONE’s publication criteria as it currently stands. Therefore, we invite you to submit a revised version of the manuscript that addresses the points raised during the review process.

This study requires additional rigor and detail to be acceptable for publication. See additional comments below.

We look forward to receiving your revised manuscript.

Kind regards,

Ramu Anandakrishnan, Ph.D.

Academic Editor

PLOS ONE

Journal Requirements:

"Michael TC Poon is supported by Cancer Research UK Brain Tumour Centre of Excellence Award (C157/A27589)."

Additional Editor Comments:

The authors claim that preclinical models of glioblastoma do not include sufficient details to compare different studies, particularly when the results are inconsistent. Clearly different results are to be expected when different studies look at different cohorts, e.g. different molecular subtypes. The question is, are these differences clear enough in the manuscript. I am concerned that these details are indeed clearly described in their manuscripts but the authors have not read these studies in sufficient detail. See Reviewer 1 comments. The “Study characteristics” section should be expanded to describe in detail differences in cohort selection, analysis method, etc. This section should be structured help answer the above question. The authors should also suggest a set of characteristics that should be listed in the Methods section so that different studies can be easily compared. In its current form this study does little more than state the obvious, albeit with specific examples. Additional rigor and detail are required for this paper to be acceptable for publication.

Reviewers' comments:

Reviewer's Responses to Questions

**Comments to the Author**

1. Is the manuscript technically sound, and do the data support the conclusions?

Reviewer #1: Yes

Reviewer #2: Partly

Reviewer #3: Partly

2. Has the statistical analysis been performed appropriately and rigorously? 

Reviewer #1: Yes

Reviewer #2: N/A

Reviewer #3: No

3. Have the authors made all data underlying the findings in their manuscript fully available?

Reviewer #1: Yes

Reviewer #2: No

Reviewer #3: No

4. Is the manuscript presented in an intelligible fashion and written in standard English?

Reviewer #1: Yes

Reviewer #2: Yes

Reviewer #3: Yes

5. Review Comments to the Author

Reviewer #1: The authors reasoned that preclinical models for association testing between molecular markers and overall patient survival should include more complete patient characteristics. This was studied in the TCGA glioblastoma cohorts in RNA microarray data. They concluded that preclinical GBM studies with incomplete patient characterization resulted in inconsistent results.

Comments:

1. CCGA should be mentioned in the abstract. Also, RNA sequencing and RNA microarrays were assessed in this analysis.

2. The results from Table 2 should provide context and be modified e.g. EGFR was reported to not have inconsistent results because Kuang et al. reported a Kaplan-Meier survival analysis using data from The Cancer Genome Atlas dataset which indicated that the expression of Cx43 significantly improved the prognosis of GBM patients who express EGFR while Li et al., showed that neither EGFR nor NTN4 expression significantly correlated with patient survival after TMZ treatment, while co-expression of EGFR/NTN4 predicts poor patient survival. These two studies can not be compared because they are looking at specific subgroups within their pool of patients which was defined by other molecular markers.

Also, Genovese et al. integrated data to produce a network model with molecular subtypes of GBM and functional genomic screen to determine associations with patient survival. Each pool of patients is sub categorized into specific glioblastoma molecular subtypes.

The authors make a valid and important point that translational studies in GBM should increase their transparency and consider patient characteristics when publishing their findings. I suggest that the authors modify their results after reviewing the manuscripts in Table 2.

Reviewer #2: In ' Analytic approaches to clinical validation of results from preclinical models of glioblastoma: a systematic review,' the authors searched on Embase and Medline for glioblastoma studies using preclinical models that also utilized data from TCGA and/or CGGA to validate the association between molecular markers and overall survival. Out of the 58 eligible studies from 1,751 non-duplicate records, they sorted out a total of 126 molecular markers that were reported. And of the 15 molecular markers reported and analyzed in more than one study, five showed discrepancies among studies. The authors argued that these inconsistent results are probably due to different analytic approaches of survival analyses used in different studies, thus concluded that increasing the transparency in reporting and the use of analytic methods that adjust for clinical variables could improve the reproducibility of these findings. This is a short but interesting study, even though the finding – discrepancies among studies – is not surprising. The authors discussed a few possibilities to improve the reproducibility of studies, such as using multivariable analysis considering multiple clinical variables, providing the query data and code-based commands used in the study. I am not sure why the authors picked glioblastoma. However, suppose they can expand the review on other cancer types with data available from TCGA. In that case, it might be as well useful to discuss about how to filter out studies with unreliable results.

Reviewer #3: The author have done a commendable work of selecting relevant studies from a large corpus to assess consistency and proper utilization of patient characteristics in those studies.

A brief review/overview of survival analysis with different variables can set the proper context for this review work. This review paper is missing the necessary background. The results are not explicitly tied to the studies under review. The authors should state the findings from the studies while presenting their conclusion.

Other comments:

Move the citation number before the full stop throughout the document. ...developing novel therapies.[5] >> ...developing novel therapies[5].

The authors have chosen studies for their review that included data from TCGA and CGGA but haven't considered ICGA. Is there a reason for omitting the last repository?

It was not clear what the authors meant by "stratified by the data sources" on page 5, the first paragraph. A structured demonstration of stratified results would be helpful.

The authors haven't listed the screening process in detail. I would love to know the different criteria the first two reviewers considered for accepting or rejecting a paper. This will also give more credibility to the selected studies.

Please cite Covidence. "A set of molecular markers was defined by a grouping..." -- what is this grouping and what is the process and sigficance of this grouping?

Page 6, "Quality assessment", please write the first sentence to clarify. Please quantify/quantitatively define the measures of quality relating to the risk of bias.

The authors claimed the multivariable analyses are better than univariable. However, there is no direct proof to support the claim in this context.

Page 7, Reproducibility section: It's unclear why the 12 studies that did not perform a multivariable survival analysis should have the same number of patients included.

The authors should demonstrate the concept of consistency between studies in terms of some biomarkers using examples and quantifiable/qualifiable metrics. Also, what is the impact of these inconsistencies? Is there a way to judge which study to trust in case of such inconsistency?

6. PLOS authors have the option to publish the peer review history of their article (what does this mean?). If published, this will include your full peer review and any attached files.

Reviewer #1: No

Reviewer #2: No

Reviewer #3: No

---

## [Author Response · Author response to Decision Letter 0]

20 Jan 2022

Please refer to the "Response to Reviewers" document.

---

## [Editor Report · Decision Letter 1]

16 Feb 2022

Analytic approaches to clinical validation of results from preclinical models of glioblastoma: a systematic review

PONE-D-21-32960R1

Dear Dr. Poon,

We’re pleased to inform you that your manuscript has been judged scientifically suitable for publication and will be formally accepted for publication once it meets all outstanding technical requirements.

Kind regards,

Ramu Anandakrishnan, Ph.D.

Academic Editor

PLOS ONE
---

## [Editor Report · Acceptance letter]

21 Feb 2022

PONE-D-21-32960R1 

Analytic approaches to clinical validation of results from preclinical models of glioblastoma: a systematic review 

Dear Dr. Poon:

I'm pleased to inform you that your manuscript has been deemed suitable for publication in PLOS ONE. Congratulations! Your manuscript is now with our production department. 

Kind regards, 

on behalf of

Dr. Ramu Anandakrishnan 

Academic Editor

PLOS ONE